# Negation of Pythagorean Fuzzy Number Based on a New Uncertainty Measure Applied in a Service Supplier Selection System

**DOI:** 10.3390/e22020195

**Published:** 2020-02-07

**Authors:** Haiyi Mao, Rui Cai

**Affiliations:** 1School of Computer and Information Science, Southwest University, No. 2 Tiansheng Road, BeiBei District, Chongqing 400715, China; 2Business College, Southwest University, No.160 Xueyuan Road, Rongchang District, Chongqing 402460, China

**Keywords:** Pythagorean fuzzy number, negation, uncertainty measure, Topsis, relative closeness index, service supplier selection system

## Abstract

The Pythagorean fuzzy number (PFN) consists of membership and non-membership as an extension of the intuitionistic fuzzy number. PFN has a larger ambiguity, and it has a stronger ability to express uncertainty. In the multi-criteria decision-making (MCDM) problem, it is also very difficult to measure the ambiguity degree of a set of PFN. A new entropy of PFN is proposed based on a technique for order of preference by similarity to ideal solution (Topsis) method of revised relative closeness index in this paper. To verify the new entropy with a good performance in uncertainty measure, a new Pythagorean fuzzy number negation approach is proposed. We develop the PFN negation and find the correlation of the uncertainty measure. Existing methods can only evaluate the ambiguity of a single PFN. The newly proposed method is suitable to systematically evaluate the uncertainty of PFN in Topsis. Nowadays, there are no uniform criteria for measuring service quality. It brings challenges to the future development of airlines. Therefore, grasping the future market trends leads to winning with advanced and high-quality services. Afterward, the applicability in the service supplier selection system with the new entropy is discussed to evaluate the service quality and measure uncertainty. Finally, the new PFN entropy is verified with a good ability in the last MCDM numerical example.

## 1. Introduction

Multi-criteria and multi-sensor information fusion [1] is now an important course. Information fusion is still an open-world issue. In the real world, people’s judgments on the possibility and uncertainty of events are mostly based on probability theory [2]. It is one of the most basic tools for dealing with uncertainty. Dempster and Shafer’s evidence theory (D–S theory) [3,4], taking advantage of basic probability assignment (BPA), has the ability to directly express uncertainty in response to conflicts [5]. In addition, a number of approaches have been proposed to settle this issue, such as fuzzy sets theory [6], and soft sets [7], entropy [8], Z-number [9], and D-number [10]. Thanks to the high efficiency, it is widespread used in multiple fields, such as data classifier [11], pattern classifier [12,13], decision-making [14,15], prediction [16,17], medical diagnosis [18,19], and so on [20]. More detailed examples are used to explain the application of evidence theory. Based on the D-number theory, a new method for evaluating GSCM practice in a fuzzy environment is proposed by Deng [21]. The main contributions and advantages of the proposed method are the ability to cope with the ambiguity, ambiguity, and non-exclusiveness in evaluating GSCM practices. Yager proposed that the use of OWA measures can provide OWA weights, which can flexibly calculate the likelihood function of probability evidence in the context of forensic crime investigations, so that optimistic or pessimistic likelihood results can be obtained [22].

The Pythagorean fuzzy number (PFS) [23] is an evolution of the intuitionistic fuzzy number (IFN) [24,25]. Fan further developed an intuitionistic fuzzy hybrid weighted arithmetic averaging (IFHWAA) operator, and an intuitionistic fuzzy hybrid weighted geometric averaging (IFHWGA) operator to solve multi-criteria decision making under an intuitionistic fuzzy environment [26]. The BPA negation has multiple approaches to measuring the ambiguity degree. It reflects the ambiguity degree of belief mass [27] how much a proposition owns, determined by the difference between the BPA negation and the BPA itself. Whether it is the negation of the Bayesian model or negation of the BPA, the negation is calculated based on an exhaustive frame of discrimination (FOD) [28]. Moreover, the results of these models after negation are also specific values. However, a specific value is far from an interval with a more powerful ability to reflect his ambiguity. Moreover, both probability and BPA cannot reflect the rejection of a proposition. Then, it is indeed of vital significance to find their negation to obtain the uncertainty measure [29]. However, the PFN itself consists of the membership and non-membership of an alternative. The PFN ambiguity degree is determined by both alternate membership and non-membership. Research on the negation of PFN has more important practical significance. Nowadays, there is no uniform standard for the measurement of service quality and a lack of a scientific and systematic evaluation system. It brings challenges to the future development of airlines.

A new method of uncertainty measures based on a new Pythagorean fuzzy number entropy is proposed in this paper. PFN negation is an essential tool in MCDM problems to express the degree of fuzziness for alternatives. Furthermore, to verify the new entropy, a Pythagorean fuzzy number negation approach is proposed and finds the correlation of the uncertainty measure. The new method is helpful for airlines to establish a reasonable service quality evaluation system and help enterprises improve their competitiveness in service quality. Finally, the new PFN entropy is verified to own a strong ability of PFN uncertainty measure.

In Section 2.1, some basic properties of PFN and various distance measure definitions are reviewed. In Section 2.2, negations based on probability distribution and BPA are reviewed. In Section 2.3, a variety of uncertainty measures are reviewed. In Section 2.4, a traditional similarity-measure-based PFN entropy and an inclusion measure method will be reviewed. In Section 3.1, a new entropy based on the Pythagorean fuzzy number for uncertainty measure is proposed. In Section 3.2, a new Pythagorean fuzzy number negation approach is proposed. In Section 4.1 and Section 4.2, a single-criterion decision-making numerical example and a multi-criteria decision-making numerical example are listed. Moreover, the newly proposed PFN entropy is verified its rationality through the uncertainty measures. In Section 4.3, the applicability of the PFN negation approach under multi-criteria and the defect of this entropy under single-criterion have been discussed. In Section 4.4, the intuitive physical meaning of the proposed PFN entropy is discussed. In Section 5, all the proposed approaches presented in this paper are summarized.

## 2. Literature Review

### 2.1. Pythagorean Fuzzy Number

Pythagorean fuzzy number is abbreviated to PFN. Since PFS is presented, it has been widely used in many fields such as decision-making data fusion and supplier selection [30]. They both belong to the research range of the fuzzy set theory. Yager uses geometric mean and ordered weighted geometry (OWG) operators to summarize standard satisfaction [31]. Furthermore, Pythagoras membership is extended to complex numbers to effectively express uncertainty [32]. Han proposed an interval-valued Pythagorean prioritized operator-based game theoretical framework to mitigate the cross-influence problem [33]. IFN $\left(x_{\beta },y_{\beta }\right)$ represents a confidence interval and $x_{\beta }+y_{\beta }\leq 1$, where $x_{\beta }$ and $y_{\beta }$ represent a minimum membership degree and a maximum membership degree. Feng proposes several compatible properties of preorders concerning the algebraic sum and scalar product operations of IFN, and apply them to the investigation of various lexicographic orders [34]. Song conducted medical diagnosis and cluster analysis experiments to illustrate the applicability of the proposed IFN similarity measure in practice [35]. The PFS $\left(u_{p},v_{p}\right)$ whose elements $u_{p}$ and $v_{p}$ represent the membership and non-membership. It is necessary to satisfy that $u_{p}^{2}+v_{p}^{2}\leq 1$. Obviously, PFS can express more fuzzy information than IFS because it has more space to store information than IFS. It should be noted that an IFN must be a PFN. Xiao applies PFN to medical diagnosis [36]. There is a strong advantage in solving multi-criteria decision-making problems (MCDM) problems with PFN. Fei considers the credibility degree and applies PFN to the risk assessment problem [37].

PFS $\left(u_{p},v_{p}\right)$ is used to characterize the membership and non-membership of a proposition. $\pi_{p}^{2}$ is denoted as the degree of indeterminacy, called hesitancy function, satisfying
(1)$$\pi_{p}^{2}=1-u_{p}^{2}-v_{p}^{2}$$

There are three basic forms of PFN proposed by Yager [38]. He called the fuzzy subset with these Pythagorean status levels PFS. They are represented by ordered pairs $\left(u_{p},v_{p}\right)$, polar coordinates $\left(r_{p},\theta \right)$ and ordered pairs $\left(r_{p},d\right)$ [39].

In the first two forms:(2)$$r_{p}^{2}=u_{p}^{2}+v_{p}^{2}$$
(3)$$\theta =tan^{-1}\left(\frac{v_{p}}{u_{p}}\right)$$

In the third form:(4)$$d=1-\frac{2\theta }{\pi }$$

The extraordinary advantage of PFN $\left(u_{p},v_{p}\right)$ is that it not only allows IFN $\left(u_{\beta },v_{\beta }\right)$ to capture the sum of both the degrees provided by decision-makers. In the case where the alternative does not satisfy the sum of IFN less than or equal to one, this is
(5)$$u_{\beta }+v_{\beta }\leq 1$$

It can still be modeled, in place of other cases that IFN cannot describe, as long as its sum of squares is less than or equal to one. However, this advantage of PFN is at the expense of more complex operations. Usually, PFN (1,0) is used to represent $\sigma^{+}$, and its meaning represents full trust to the proposition. PFN (0,1) represents $\sigma^{-}$, and its meaning represents a complete denial of the proposition.

There are a series of approaches to expressing belief in PFN.

#### 2.1.1. Score Function

The score function (S) is a standard used to characterize the belief mass in a proposition [40]:(6)$$S=u_{p}^{2}-v_{p}^{2}$$

Its accuracy function (H) [41] is defined as
(7)$$H=u_{p}^{2}+v_{p}^{2}$$

An approach to comparing belief masses of two PFN $x_{1}$ and $x_{2}$ are shown as follows:if S($x_{1}$) > S($x_{2}$), then $x_{1}$ > $x_{2}$.if S($x_{1}$) < S($x_{2}$), then $x_{1}$ < $x_{2}$.if S($x_{1}$) = S($x_{2}$) and H($x_{1}$) > S($x_{2}$), then $x_{1}$ > $x_{2}$.if S($x_{1}$) = S($x_{2}$) and H($x_{1}$) < S($x_{2}$), then $x_{1}$ < $x_{2}$.if S($x_{1}$) = S($x_{2}$) and H($x_{1}$) = S($x_{2}$), then $x_{1}$ = $x_{2}$.

#### 2.1.2. Relative Closeness (*R*) and Revised Relative Closeness Index (ζ)

Relative closeness index (*R*) and revised relative closeness index (ζ) are two methods to represent belief mass that a proposition gains [39]. The ambiguity can also be measured by the relative closeness index (*R*) as well. The Hamming distance (D) [42] between two independent propositions is defined as
(8)$$D\left(x_{i},x_{j}\right)=\frac{1}{2}\times \left(\right|u_{i}^{2}-u_{j}^{2}\left|+\right|v_{i}^{2}-v_{j}^{2}\left|+\right|\pi_{i}^{2}-\pi_{j}^{2}\left|\right)$$

Relative closeness index (*R*) is defined as
(9)$$R\left(x_{i}\right)=\frac{D(x_{i},\sigma^{-})}{D\left(x_{i},\sigma^{-}\right)+D\left(x_{i},\sigma^{+}\right)}$$

Score function (S), relative closeness index (*R*), and accuracy function (H) are drawn in Figure 1 all over PFN domain. Co-domains of score function (S), relative closeness index (*R*), and accuracy function (H) are defined as
(10)$$\;S\left(x_{i}\right)\;\leftarrow \;\left[-1,\;1\right]$$
(11)$$\;R\left(x_{i}\right)\;\leftarrow \;\left[0,\;1\right]$$
(12)$$\;H\left(x_{i}\right)\;\leftarrow \;\left[0,\;1\right]$$

Technique for Order of Preference by Similarity to Ideal Solution (TOPSIS) is a multi-criteria decision-making analysis method originally developed by Hwang and Yoon [43]. The chosen alternative should be the shortest geometric distance from the positive ideal solution (PIS) and the longest geometric distance from the negative ideal solution (NIS). It is a method of compensatory aggregation, which identifies the weight of each criterion. Moreover, it compares the geometric distance between alternative and ideal alternative. Revised relative closeness index (ζ) compares the geometric distance between alternative and relative ideal alternative. It is an optimization based on the relative closeness index (R) in TOPSIS. Compensation methods such as TOPSIS allow trade-offs between multiple criteria. Poor results from one criterion can be offset by good results from the others. This provides a more realistic form of modeling than non-compensated methods.

The distance between an alternative and the negative ideal solution (NIS) can measure the degree of trust of an alternative in TOPSIS. In addition, the distance between an alternative and the positive ideal solution (PIS) is distrust degree of an alternative in TOPSIS [39]. Assume that, in an MCDM problem, there are two ways to generate the PIS and NIS. One is to directly define the absolute PIS (1,0) of PFN as $\sigma^{+}$, and directly define the absolute NIS (0,1) of PFN as $\sigma^{-}$. Since the maximum distance of the two alternatives $x_{i}$ and $x_{j}$ is within the range of $\theta \in [0,\frac{\pi }{2}]$, $r_{p}\in \left[0,1\right]$. That is,
(13)$$D\left(x_{i},x_{j}\right)\leq \sqrt{2}$$

The calculation of each distance to PIS or NIS is performed as follows:(14)$$D\left(x_{i},\sigma^{+}\right)=\frac{1}{2}\times \left(1-u_{p}^{2}+v_{p}^{2}+\pi_{p}^{2}\right)=1-u_{p}^{2}$$
(15)$$D\left(x_{i},\sigma^{-}\right)=\frac{1}{2}\times \left(u_{p}^{2}+1-v_{p}^{2}+\pi_{p}^{2}\right)=1-v_{p}^{2}$$

Revised relative closeness index (ζ) is another method in a specific MCDM proposed by Hadi-Vencheh and Mirjaberi [44]. Maximum and minimum score function (S) could be obtained from multiple alternatives in the same criterion. The maximum value is the relatively optimal one among all alternatives as PIS of the PFN marked as $x^{+}$. On the contrary, the minimum value is relatively negative one as NIS PFN marked as $x^{-}$.

However, Hadi-Vencheh shows that, in some cases, the relative closeness index (ζ) cannot achieve the best alternative which has the shortest distance from the PIS and the farthest distance from the NIS with a TOPSIS method in MCDM. Therefore, he suggests that people can use the following formula, called revised relative closeness index (ζ), instead of the relative closeness index (*R*):(16)$$\zeta =\frac{D(x_{i},x^{-})}{D_{max}\left(x_{i},x^{-}\right)}-\frac{D(x_{i},x^{+})}{D_{min}\left(x_{i},x^{+}\right)}$$

When cardinality of PFN is *n*, $D_{max}\left(x_{i},x^{-}\right)$ and $D_{min}\left(x_{i},x^{+}\right)$ are defined as
(17)$$D_{max}\left(x_{i},x^{-}\right)=\max_{1\leq i\leq n}D\left(x_{i},x^{-}\right)$$
(18)$$D_{min}\left(x_{i},x^{+}\right)=\min_{1\leq i\leq n}D\left(x_{i},x^{+}\right)$$

The distance function (D) and the closeness index between the respective alternatives are found by the two methods above. Moreover, the PFN is converted into belief mass. The belief mass function can be directly compared with other intuitively. This method is widely used, and it is the link between the PFN and D–S theory belief mass.

### 2.2. Negation of Probability Distribution and Basic Probability Assignment Methods

Yager [45] proposed a transformation to obtain the negation of the probability distribution and the definition of the negation in the probability distribution.
(19)$${\bar{p}}_{i}=\frac{1-p_{i}}{n-1}$$

This follows when $p_{i}\geq p_{j}$, then $\bar{p_{i}}\leq \bar{p_{j}}$.

The negation after twice is not equal to the original probability because it is in the form of entropy increase in negation iterations:(20)$${\bar{\bar{p}}}_{i}\neq p_{i}$$

After *j* iterations of negation, ${\bar{p}}_{j}$ satisfies
(21)$${\bar{p}}_{j}=\frac{na_{1}-1}{n{\left(n-1\right)}^{j-1}}+\frac{1}{n}$$

After infinite iterations of negation, probability distribution eventually converges to normal distribution, that is,
(22)$$\lim_{j\to \infty }{\bar{p}}_{j}=\frac{1}{n}$$

Each set whose mass function not equal to zero is defined as a focal element. Under the condition of maximum entropy, the assignment of mass function exists in a unique form. Profoundly, more new properties of probability distribution negation have been proposed by Srivastava [46].

Yin generalizes the negation of probability theory to BPA [47]. Under an exhaustive framework of discriminant (FOD), the negation of BPA can be expressed as
(23)$${\bar{m}}_{a}=\frac{1-m_{a}}{n-1}$$

Gao and Deng proposed that mass function could be assigned to the whole set except conflict set (ϕ) and the focal element itself after negation [48]:(24)$${\bar{m}}_{a}=\frac{1-m_{a}}{2^{n}-2}$$

Xie and Xiao proposed that BPA negation matrix can be established according to the mutually exclusive relationship between multiple subsets [49], to reassign mass function to the BPA negation proposition in an exhaustive FOD.

### 2.3. Uncertainty Measure Methods

Uncertainty measure plays an important role in real application [50,51].

In a cardinality equal to *n* probability distribution, Tsallis entropy is defined as
(25)$$H_{T}\left(P\right)=\frac{k}{q-1}\left(1-\sum_{i=1}^{n}p_{i}^{q}\right)$$

When *q* converges to 1, it degenerates to Bolzmann–Gibbs entropy [52] as
(26)$$H_{B}\left(P\right)=-\sum_{i=1}^{n}p_{i}\times log\left(p_{i}\right)$$

Gao proposed an uncertainty measure Based on Tsallis entropy in evidence theory [53]. Kang proposed a negation of a probability distribution considering the influence of the correlations in a system from the perspective of Tsallis entropy [54] as
(27)$${\bar{p}}_{i}=\frac{\frac{1}{k-1}\left(1-p_{i}^{k}\right)}{\sum_{x_{i}\in X}\frac{1}{k-1}\left(1-p_{i}^{k}\right)}$$

Shannon entropy is also known as information entropy [55]. The formula of Shannon entropy has a similar mathematical model to the thermal entropy in uncertainty measure of random variables. Shannon entropy is defined as
(28)$$H_{S}\left(X\right)=-\sum_{x}P\left(x\right)\times log_{2}P\left(x\right)$$

The greater uncertainty measure, the larger entropy, and the greater amount of information it contains. Gini entropy has no complicated logarithmic operation log, which greatly simplifies the computational complexity without losing much information [56]:(29)$$\begin{array}{cc} H_{G}\left(X\right) & =-\sum_{x}P\left(x\right)\times \left(1-P\left(x\right)\right)\\ \; & =1-\sum_{x}P{\left(x\right)}^{2} \end{array}$$

Both Shannon entropy and the GINI entropy have a maximum entropy when they are normally distributed.

Uncertainty measures are widely used in life. Zhou proposed an improved algorithm using PFSDM distance in medical diagnosis, which can avoid counter-intuitive results, especially when there are data conflicts [57]. Gao uses a Pseudo-Pascal Triangle to explore the physical meaning of Deng Entropy [58]. On this basis, Liu proposed generalized Rényi–Deng (R–D) and Tsallis–Deng (T–D) entropy to control conflicts [59]. Li proposed that the evidence decision tree can be directly applied to classification, which effectively reduces the complexity of the algorithm [60]. Wen uses entropy to study the vulnerability of complex networks [61].

The uncertainty measurement method plays an important role in network nodes and power systems [62]. Rong proposed a method of reliability analysis, which can be applied to practical projects with more fault data [63]. Li proposed the use of uncertainty measure theory for multi-state systems to calculate uncertain shocks, which can be applied to non-fatal and catastrophic failures of machines [64]. Wang applied uncertainty theory to analyze the fault characteristics and stability of communication nodes [65]. In addition, he proposed a new framework for modeling fault propagation paths of power systems based on the Ev-SNP system [66]. Liu uses uncertainty metrics to diagnose grid faults [67].

### 2.4. Pythagorean Fuzzy Number Entropy

Generally, the calculation of PFN entropy requires that converting the PFN distance function to a similarity measure. Then, the ambiguity degree of PFN is represented by the similarity measure between them. There was originally a simple approach to directly converting similarity measures into PFN entropy. Supposed that there is a Pythagorean fuzzy number (M), PFN entropy (E), and similarity measure (S) of PFN, respectively, that satisfies
(30)$$S(M,N)=1-\frac{1}{2|X|}\times \sum_{x\in X}\left(\right|u_{M}^{2}\left(x\right)-u_{N}^{2}\left(x\right)|\;+|v_{M}^{2}\left(x\right)-v_{N}^{2}\left(x\right)|+|\pi_{M}^{2}\left(x\right)-\pi_{N}^{2}\left(x\right)\left|\right)$$
(31)$$E\left(M\right)=S\left(M,M^{C}\right)=1-D\left(M,M^{C}\right)$$
where E(M) is the PFN entropy of *M*, |X| is the cardinality of criteria in the multi-criteria decision-making problem, and the focal element *x* is a multi-subset.

However, when the definite value of their Euclidean distance is extremely small, it does not mean that the ambiguity degree of the information, which is represented by the PFN membership and non-membership degree, is also very low. On the contrary, to some extent, they have a large order of magnitude difference. The similarity measure and PFN entropy will be affected by the extreme minimum value, which might lead to an inaccurate ambiguity of PFN. For example, there is a couple of PFN as an example:$$u_{M}^{2}=10^{-6},v_{M}^{2}=10^{-3},\pi_{M}^{2}=1.0000$$
$$u_{N}^{2}=10^{-2},v_{N}^{2}=10^{-3},\pi_{N}^{2}=0.9999$$

Therefore, we have similarity measure (S) and PFN entropy (E) from Equations (Equation 30) and (Equation 31)
S=0.999850
E=1−S=0.000150

This is not intuitive. They have large differences in membership, but they cannot be clearly expressed ambiguity degree with distance functions of Equation (Equation 30). The newly proposed PFN entropy in this paper will improve this deficiency.

Peng proposed a new PFN entropy based on inclusion measures and applied it in the medical field [68]. The new PFN entropy calculation approach to converting inclusion measure (I) to similarity measure (S). Finally, PFN entropy (E) is derived:(32)$$\begin{array}{cc} I(M,N) & =1-\frac{1}{2|X|}\times \sum_{x\in X}\left(\right|u_{M}^{2}\left(x\right)-u_{M}^{2}\left(x\right)\wedge u_{N}^{2}\left(x\right)|\\ \; & +|v_{M}^{2}\left(x\right)-v_{M}^{2}\left(x\right)\vee v_{N}^{2}\left(x\right)\left|\right) \end{array}$$
(33)I(M,N)=S(M,M∩N)=1−D(M,M∩N)

Inclusion measure and similarity measure based on three PFNs $X=\left\{M,N,O\right\}$ satisfy that

**Proposition** **1.**
*If M⊂N, then I(M,N)=S(M,M∩N)=1*


**Proposition** **2.**
*If I(M,N)=0, then S(M,M∩N)=0 and M=θ,N=ϕ*


**Proposition** **3.**
*If M⊂N⊂O, then I(O,M)≤S(N,M) and I(O,M)≤S(O,N)*


It is applicable when the alternative is a multi-subset. The new PFN entropy is affected by both the cardinality of focal elements and the Pythagorean fuzzy number at the same time. Furthermore, the signal-to-noise ratio is higher.

## 3. New Pythagorean Fuzzy Number Negation and Entropy

### 3.1. Pythagorean Fuzzy Number Entropy

Entropy is an indicator of energy degradation. Entropy is also used to calculate the disordered phenomenon in a system, that is, to calculate the degree of ambiguity of the system. The greater the entropy, the more information is contained. Pythagorean fuzzy number entropy is proposed to measure the degree of ambiguity of PFN in this paper. The ambiguity of the PFN is determined by the combination of the belief mass and accuracy.

On the one hand, the score function (S), the relative closeness index (*R*), and the revised relative closeness index (ζ) are all able to represent the belief mass. However, the score function (S) oscillates around zero in PFN negation. The revised relative closeness index (ζ) is exactly in the form of a negative number, and the largest ζ is equal to zero.

On the other hand, on the condition that the two PFNs have the same belief mass, the accuracy function (H) can be more intuitively proposed that the larger the accuracy function (H), the less PFN ambiguity. Therefore, the accuracy function (H) plays an extremely important role in PFN entropy.

Based on accuracy function (H), the score function (S), revised relative closeness index (ζ), and the basic idea of Shannon entropy, PFN entropy with a TOPSIS method is proposed as
(34)$$\begin{array}{c} E_{\zeta }=-\sum_{i=1}^{n}(|S\left(x_{i}\right)\times \zeta \left(x_{i}\right)\left|+H\left(x_{i}\right)\right)\\ \times log(|S\left(x_{i}\right)\times \zeta \left(x_{i}\right)|+H\left(x_{i}\right)) \end{array}$$

The PFN entropy must satisfy
(35)$$\sum_{i=1}^{n}(\;|S\left(x_{i}\right)\times \zeta \left(x_{i}\right)\left|+H\left(x_{i}\right)\;\right)=1$$

That is, it is necessary to first ensure that the above parameters of the PFN are orthogonalized before PFN entropy calculation.

### 3.2. Definition of Pythagorean Fuzzy Number Negation

The PFN consists of membership and non-membership of an alternative. It contains more information in the MCDM problem and has a larger advantage. Typically, the score function (S) is defined to characterize his belief mass. The higher the score, the higher the belief mass in this alternative directly. The ambiguity of the alternative is represented by the relative closeness index (*R*). When the relative closeness index (*R*) approaches $\frac{1}{2}$, the data is the fuzziest, displaying the most ambiguous situation. Likely, the relative closeness index (*R*) can reflect the belief mass of PFN as well. The larger magnitude of *R* represented more belief mass.

This paper proposes a negation of PFN, which intends to use the PFN negation to characterize the degree of ambiguity in the trust of an alternative. Cardinality of alternatives is *n* for an MCDM problem. Under the circumstance that only one criterion exists, the PFN is expressed as
$$\left\{\left(u_{1},v_{1}\right),\left(u_{2},v_{2}\right)\cdots \left(u_{i},v_{i}\right)\cdots \left(u_{n},v_{n}\right)\right\}$$

They all satisfy
(36)$$u_{i}^{2}+v_{i}^{2}+\pi_{i}^{2}=1$$
where i=1,2,⋯,n.

A new Pythagorean fuzzy number negation is proposed that
(37)$$\bar{u_{i}}=\frac{1-u_{i}^{2}}{n-1}$$
(38)$$\bar{v_{i}}=\frac{1-v_{i}^{2}}{n-1}$$
(39)$${\bar{\pi_{i}}}^{2}=1-{\bar{u_{i}}}^{2}-{\bar{v_{i}}}^{2}$$

The new PFN ($\bar{u_{i}}$, $\bar{v_{i}}$) is Pythagorean fuzzy number negation of the original one ($u_{i}$, $v_{i}$).

Define that the convergence value of $u_{i}$ is $u_{conv}$. Assuming multiple iterations of the negation, a convergence equation for $u_{i}$ can be obtained. The following proofs apply equally to $u_{i}$ and $v_{i}$, taking $u_{i}$ as an example as follows:(40)$$u_{conv}=\frac{1-u_{conv}^{2}}{n-1}$$

**Proposition** **4.**
*When n increases, $u_{conv}$ monotonically decreases (n≥2).*


**Proof.** Supposed that there is Proposition 5.

**Proposition** **5.**
*When $u_{conv}$ is incremented, n monotonically decreases.*
Then, Propositions 4 and 5 are mutually opposite propositions because
(41)$$\frac{1}{u_{conv}}-u_{conv}=n-1$$Obviously, when $u_{conv}$ is decremented, *n* monotonically increases. Proposition 5 is a true proposition. Since Propositions 4 and 5 are mutually opposite propositions, Proposition 4 is true as well. This is proven. □

When n=2 is, $u_{conv}$ takes the maximum value.
(42)$$\max\left(u_{conv}\right)=\frac{\sqrt{5}-1}{2}$$

Since $u_{i}$ and $v_{i}$ converge toward the same direction, as long as the convergence value of $u_{conv}$ satisfies
(43)$$\max\left(u_{conv}\right)\leq \sqrt{\frac{1}{2}}$$

Then, the Pythagorean fuzzy number negation satisfies the definition of the PFN. After multiple PFN negation iterations, $u_{i}$ will converge to $u_{conv}$ in Figure 2:(44)$$u_{conv}=\frac{\left(1-n\right)+\sqrt{{\left(n-1\right)}^{2}+4}}{2}$$

The newly proposed PFN negation operation is used to verify the properties of the newly proposed PFN entropy in the previous section.

**Proposition** **6.**
*During the negation operation, although PFN entropy is not monotonically increasing, the overall trend is that, as the number of iterations increases, PFN entropy increases.*


**Proposition** **7.**
*The above PFN negation operation satisfies the maximum distribution of entropy after infinite iterations; meanwhile, membership ($u_{conv}$) and non-membership ($v_{conv}$) satisfy the maximum distribution of entropy.*


Establishing an evaluation system of the PFN ambiguity degree is of vital significance for the service supplier selection system. PFN entropy obtains an uncertain measure of the entire supplier system by comprehensively evaluating the support of each supplier. Usually, the result of the service supplier selection problem is to choose the optimal alternative. Uncertainty measurement will target all service suppliers and propose the gap between the optimal solution and other alternatives. Therefore, studying PFN entropy is an important course of studying the reliability of the optimal alternatives in supplier selection.

Besides, the corresponding formal algorithm is given in Algorithm 1, flowchart is given in Figure 3. Some numerical examples are shown below.

 **Algorithm 1:** An algorithm of PFN negation based on new entropy  **Require:** PFN in MCDM, *n* alternatives, *m* criteria and weighted factors *w*  **Ensure:** Scores function (S), accuracy function (H), (revised) relative closeness index (*R*) and (*ζ*)  
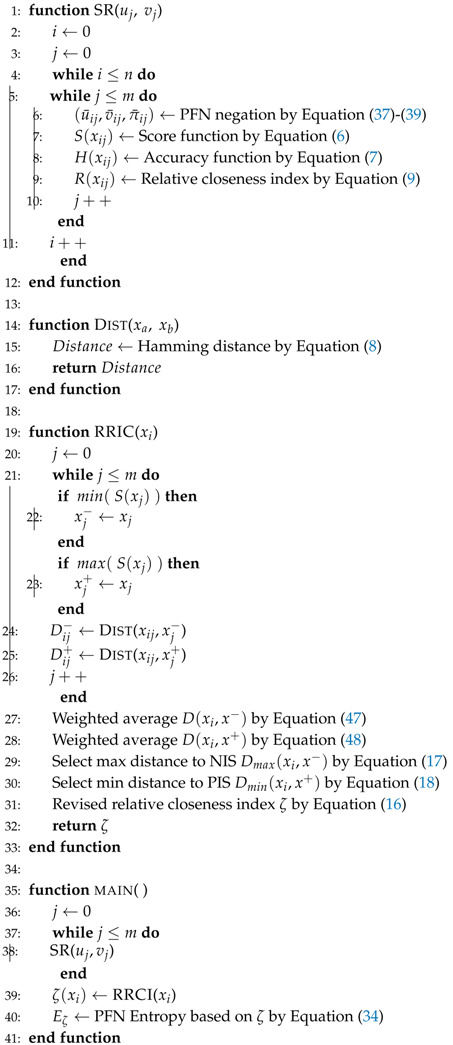


## 4. Numerical Examples and Discussion

### 4.1. Numerical Example 1

Suppose there is a series of alternatives for *A*, *B*, and *C*. The PFN under the same criterion are listed in Table 1:$$A\left(u_{A},v_{A}\right)=\left(0.9,0.3\right)$$
$$B\left(u_{B},v_{B}\right)=\left(0.4,0.7\right)$$
$$C\left(u_{C},v_{C}\right)=\left(0.8,0.4\right)$$

The example will solve the single-criterion decision-making, which is simplified by Multi-criteria decision-making (MCDM), under the same criterion with the method of PFN negation based on the relative closeness index (*R*).

Solution to Numerical Example 1:

After nine negation iterations, the PFN negation is $\left(\bar{u_{A}},\bar{v_{A}}\right)$, $\left(\bar{u_{B}},\bar{v_{B}}\right)$, $\left(\bar{u_{C}},\bar{v_{C}}\right)$. Their score function (H) and relative closeness index (*R*) are listed in the Table 2, Table 3 and Table 4 as follow.

The relative closeness index (*R*) directly reflects the degree of ambiguity about data. When it is at the process of multiple negation iterations, both $u_{i}$ and $v_{i}$ converge to $u_{conv}$ and $v_{conv}$. The relative closeness index (*R*) oscillates at $\frac{1}{2}$ up and down and finally converges to $\frac{1}{2}$. The score function (S) and the relative closeness index (*R*) of the three PFN under the same criterion are simultaneously drawn in Figure 4.

The score function (S), NIS, and PIS among the PFN in the nine iterations are listed in Table 5.

The distance function $D(x_{i},x^{-})$ of each alternative with NIS, the distance function $D(x_{i},x^{+})$ of each alternative, and PIS are defined as
(45)$$D\left(x_{i},x^{-}\right)=\frac{1}{2}\times (|u_{i}^{2}-{\left(u^{-}\right)}^{2}\left|+\right|v_{i}^{2}-{\left(v^{-}\right)}^{2}|\;+|\pi_{i}^{2}-{\left(\pi^{-}\right)}^{2}\left|\right)$$
(46)$$D\left(x_{i},x^{+}\right)=\frac{1}{2}\times (|u_{i}^{2}-{\left(u^{+}\right)}^{2}\left|+\right|v_{i}^{2}-{\left(v^{+}\right)}^{2}|\;+|\pi_{i}^{2}-{\left(\pi^{+}\right)}^{2}\left|\right)$$

$D(x_{i},x^{-})$, the distance between each alternative and NIS in the PFN negation iterations, are listed in Table 6. $D(x_{i},x^{+})$, the distance between each alternative and PIS in the PFN negation iterations, are listed in Table 7. It should be noted that $x^{-}$ is NIS and $x^{+}$ is PIS relying on score function (S) in Table 5. The relative closeness index (*R*) among the PFN in the nine negation iterations are listed in Table 8.

PFNs are extracted by the magnitude of belief mass under the same criterion. The most trusted alternative is obtained as PIS, and the least trusted alternative is obtained as NIS in each iteration. Decision-making is the same relying on either score function (S) or relative closeness index (*R*).

### 4.2. Numerical Example 2

In the context of global economic integration, service levels in various industries have made breakthroughs. The national economy continues to grow, and the people’s growing requirements promote the rapid development of tourism. With the rapid development of related industries, the air transport industry has been developed extremely fast. In the world’s highly competitive aviation service field, low-level competitive business methods such as ticket price reduction and cost reduction are inferior. Reducing prices and compressing costs can indeed solve the immediate need, but it will affect the normal market strategy of the enterprise. In the long run, this will cause airlines to be trapped in a difficult situation of price competition. Nowadays, there is no uniform criteria for measuring service quality, and there is no scientific and systematic measurement index system. It brings challenges to the future development of airlines. Therefore, they must grasp the future market trends and win with advanced and high-quality services. How to make a good use of characteristics of the airlines and the consumer psychology of passengers to establish a reasonable service quality evaluation system to help companies improve the competitiveness of service quality companies has become the main content of research. It has practical significance.

As for a service supplier selection MCDM problem from Ref. [39], three major airlines (UNI Air ($x_{A}$), Transasia ($x_{B}$), Mandarin ($x_{C}$)) determine which is the optimal service supplier option through four criteria (Booking and ticketing service ($C_{1}$), Check-in and boarding process ($C_{2}$), Cabin service ($C_{3}$), and Responsiveness ($C_{4}$)). Under four criteria, cardinality *n* of alternatives is three, with ordered weighted average (OWA) [69,70] $w_{j}=\left(0.15,0.25,0.35,0.25\right)$. All data of PFNs are exhibited in Table 9. Aim to calculate PFN negation based on revised relative closeness index (ζ).

First of all, distance of PFN with the method mentioned in Section 2.4 by Equations (Equation 30) and (Equation 31) is displayed in Table 10. The total uncertainty of each alternative is shown in Table 10 after every negation iteration. The uncertainty measure of each alternative is plotted in Figure 5. In Equation (Equation 30), $\frac{1}{\left|X\right|}$ is transformed into $w_{i}$ (PFWA) operation instead. Unfortunately, the uncertainty measure of the whole service supplier selection system cannot be evaluated. The newly proposed method effectively solves this problem.

With the same method as numerical example 1 under each criterion, similar to Table 6 and Table 7, alternative A, alternative B, and alternative C each has another two independent tables distance functions $D(x_{ij},x_{j}^{+})$ and $D_{j}\left(x_{ij},x_{j}^{+}\right)$ under each new criterion, where *j* is the label of criterion, and the cardinality of criteria is *m*. With the following Equations (Equation 47) and (Equation 48), named Pythagorean fuzzy weighted aggregation (PFWA) proposed by Yager [71], distances of PFN in each criterion 2,3,4 are drawn in Figure 6, Figure 7 and Figure 8 as histograms:(47)$$D\left(x_{i},x^{-}\right)=\frac{1}{2}\sum_{j=1}^{m}w_{j}\times D\left(x_{ij},x_{j}^{-}\right)$$
(48)$$D\left(x_{i},x^{+}\right)=\frac{1}{2}\sum_{j=1}^{m}w_{j}\times D\left(x_{ij},x_{j}^{+}\right)$$

After OWA according to its own weighting factor $w_{j}$, distance functions are obtained in Table 11 and Table 12. Meanwhile, PFWA finishes. Finally, revised relative closeness index (ζ) is obtained in Table 13 of this MCDM problem.

Entropy $E_{\zeta }$ as a measure of ambiguity degree is defined as Equation (Equation 34) based on revised relative closeness index (ζ). Entropy data are listed in Table 13 and drawn in Figure 9.

### 4.3. Characters of PFN Negation

#### 4.3.1. Advantages of the Newly Proposed Uncertainty Measure

The existing PFN uncertainty measure method is not suitable for systematic evaluation of service supplier selection system problems. This uncertainty measure is only applicable to a single PFN uncertainty measure based on the negation iterations. The newly proposed method effectively solves this problem.

This paper proposes a PFN entropy based on the relative closeness index (ζ), which is a TOPSIS method, which greatly reduces the phenomenon of intuition in the Section 2.4 example due to the small membership degree. Reasonably, the newly proposed PFN entropy in the MCDM problem takes into account the contribution of the PFN’s score function (S), accuracy function (H), and relative closeness index (ζ) to the PFN fuzziness degree.

#### 4.3.2. Defects of the Newly Proposed Uncertainty Measure

In the calculation of the uncertainty measure, since the relative closeness index (ζ) must be calculated, the premise must be satisfied that the denominators $D_{max}\left(x_{i},x^{-}\right)$ and $D_{min}\left(x_{i},x^{+}\right)$ of Equation (Equation 16) must be non-zero. If this condition cannot be met, the PFN entropy newly proposed in this paper is not applicable to calculate the ambiguity degree of PFN.

We will use the following case to illustrate the inapplicability of the newly proposed PFN entropy.

In numerical example 1, since only the criterion 1 exists, the single-criterion decision-making can transmit the case of $w_{i}$ = 1 to the MCDM problem.

As for the revised relative closeness index (ζ), $D_{max}\left(x_{i},x^{-}\right)$ is the maximum distance function all over the alternatives under the total criteria.

When frequency of iterations = 0 ($x_{i}\in \left\{x_{A},x_{B},x_{C}\right\}$),
$$D_{max}\left(x_{i},x^{-}\right)=D\left(x_{A},x^{-}\right)=0.097500$$When frequency of iterations = 1 ($x_{i}\in \left\{x_{A},x_{B},x_{C}\right\}$),
$$D_{max}\left(x_{i},x^{-}\right)=D\left(x_{B},x^{-}\right)=0.025106$$

The same as $D_{min}\left(x_{i},x^{+}\right)$ under this criterion.

When frequency of iterations = 0 ($x_{i}\in \left\{x_{A},x_{B},x_{C}\right\}$),
$$D_{min}\left(x_{i},x^{+}\right)=D\left(x_{A},x^{+}\right)=0.000000$$When frequency of iterations = 1 ($x_{i}\in \left\{x_{A},x_{B},x_{C}\right\}$),
$$D_{min}\left(x_{i},x^{+}\right)=D\left(x_{B},x^{+}\right)=0.000000$$

When
(49)$$D\left(x_{i},x^{+}\right)=0\;and\;D_{min}\left(x_{i},x^{+}\right)=0,$$
this leads to
(50)$$\zeta ⇐-\frac{D(x_{i},x^{+})}{D_{min}\left(x_{i},x^{+}\right)}=NaN$$
where NaN represents “Not a Number”.

When
(51)$$D\left(x_{i},x^{+}\right)\neq 0\;and\;D_{min}\left(x_{i},x^{+}\right)=0,$$
this leads to
(52)$$\zeta ⇐-\frac{D(x_{i},x^{+})}{D_{min}\left(x_{i},x^{+}\right)}=-\infty$$

In summary, $D_{min}\left(x_{i},x^{+}\right)=0$ exists under a single criterion, leading to a revised relative closeness index lim(ζ)=−∞ in Table 14. But in the multi-criteria multi-decision problem, this problem will be effectively solved. In the MCDM problem, when calculating $D_{max}\left(x_{i},x^{-}\right)$ and $D_{min}\left(x_{i},x^{+}\right)$, the PFN of alternatives $x_{i}$ is arithmetic weighted average according to its own weighting factor. Unlike the single-criterion multi-decision, the MCDM must first calculate the OWA under each criterion. Next, calculate the revised relative closeness index (ζ) in Topsis. Therefore, in the MCDM problem, the probability that
(53)$$\lim_{D_{min}\left(x_{i},x^{+}\right)=0}\zeta =-\infty$$
is extremely low.

### 4.4. View from Entropy

The basic idea of PFN entropy is that the magnitude of entropy is determined by the combination of belief mass and accuracy. Among them, the score function (S) is used as the weight factor of the revised relative closeness index (ζ).

Since the revised relative closeness index (ζ) is always less than or equal to zero in the PFN negation iterations, whereas score function (S) converges to zero. Meanwhile, the accuracy function (H) is dramatically reduced. Therefore, at the macro level, his degree of ambiguity is increasing in the process of PFN negation iterations. One of the great advantages of the revised relative closeness index (ζ) is that it no longer regards PFN $\sigma^{-}$(0, 1) and $\sigma^{+}$(1, 0), as a reference for NIS and PIS to calculate closeness index. In each process of calculating ζ, only the relatively worst and optimal alternatives ($x^{-}$ and $x^{+}$) under this criterion are regarded as NIS and PIS. Therefore, ζ does not converge in an exact direction in the PFN negation. Instead, ζ will periodically oscillate in Figure 10. In summary, it is indeed of vital significance to use the score function as the weighting factor of the closeness index (ζ).

## 5. Conclusions

A new Pythagorean fuzzy number negation approach based on a new entropy is proposed in this paper. New PFN entropy adopts a TOPSIS method of relative closeness index. It is a method of compensation. By measuring the Hamming distance between the alternative and the ideal solution, we get support for each alternative. In addition, poor results can be offset by good results from another criterion. It provides a more reliable modeling method for belief mass allocation. Compared to the traditional similarity measure using Euclidean distance, the relative closeness index combines the global maximum and minimum distances, which greatly improves the phenomenon of intuition owing to the effects of small membership degree. In the existing uncertainty measure, the inclusion measure requires that the focal elements of the alternative must themselves be multi-subsets. If alternatives are singletons, the inclusion measure degenerates to the traditional Euclidean distance similarity measure. The newly proposed uncertainty measure of the PFN is determined by the belief mass and accuracy. On the one hand, both the score function and the relative closeness index can characterize the belief mass of the PFN. On the other hand, the accuracy function can characterize the accuracy of the PFN. To solve the problem that the PFN distance is too small to accurately measure the uncertainty of PFN, a new PFN entropy in MCDM is proposed. It can express the uncertainty of the entire system instead of the independent PFN uncertainty measure method. The newly proposed methods are applied to service supplier selection system of airlines to help companies improve the competitiveness of service quality companies, although there is a risk of failure in obtaining the revised relative closeness index, which may lead to failure in obtaining PFN entropy. However, in the multi-criteria decision-making problem, it has been discussed that the probability of failure is extremely low. It has been proven to perform well in terms of systematic expression of uncertainty.

## 6. Further Research Opportunities

This paper proposes a new PFN entropy and proposes a new PFN negation to verify the rationality of this newly proposed PFN entropy. In the future, it will be possible to research how to apply the newly proposed PFN entropy to the measure of the degree of fuzziness of PFN. Application fields include data classifier, pattern classifier, target recognition, service supplier selection system, decision-making, prediction, data fusion, medical diagnosis, fuzzy computation by microfluidics-based hardware [72], and so on. In particular, logic gates and fuzzy Boolean logic computation can be performed by microfluidics. It provides a method for autonomous control in microfluidic-based systems. The principle is the internal digital flow control mechanism of the bubble. Uncertainty measure has broad prospects in this field. At the same time, we will also study how to establish a connection between PFN and basic probability assignment (BPA), and whether they can use a uniform entropy to measure fuzziness degree.

## Figures and Tables

**Figure 1 entropy-22-00195-f001:**
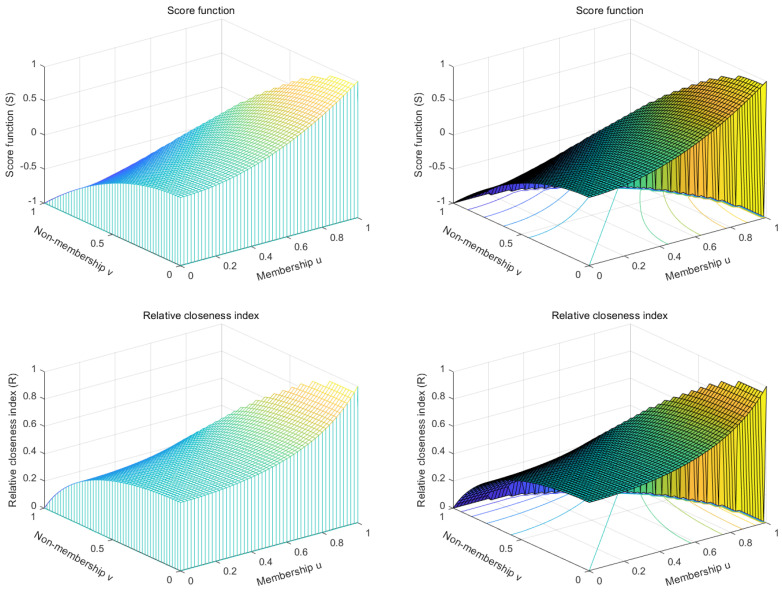
Scores function (S), relative closeness index (*R*), and accuracy function (H) in the PFN domain.

**Figure 2 entropy-22-00195-f002:**
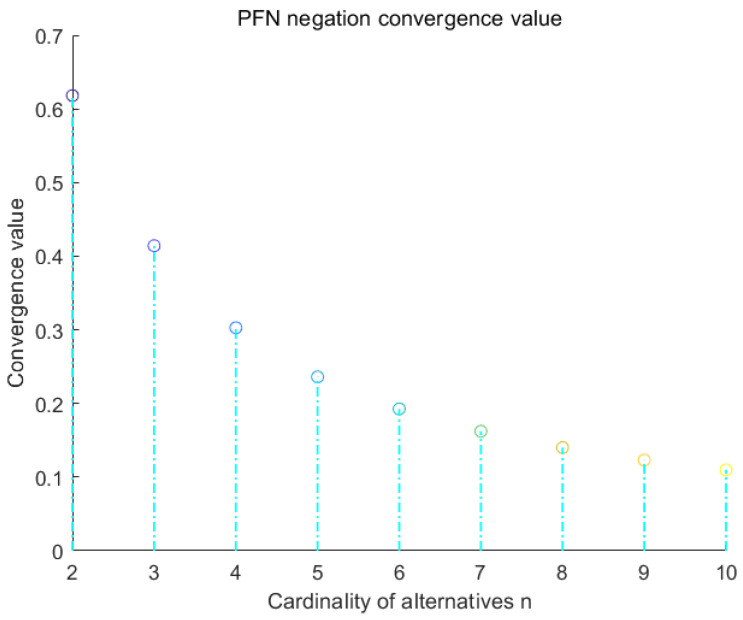
PFN negation convergence value.

**Figure 3 entropy-22-00195-f003:**
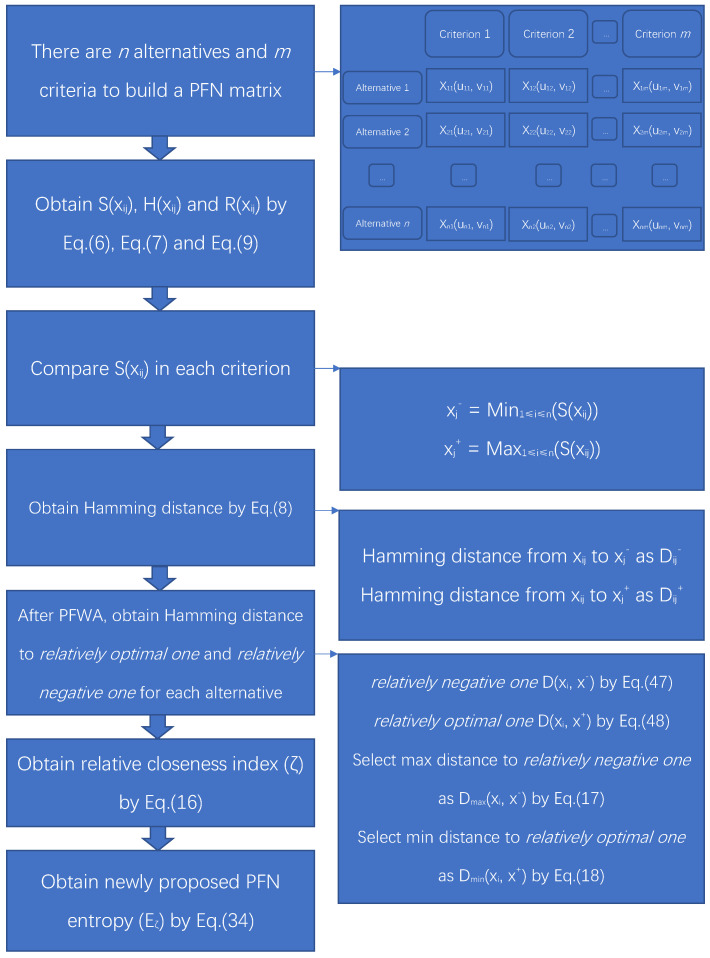
Flowchart of newly proposed PFN entropy based on the negation method.

**Figure 4 entropy-22-00195-f004:**
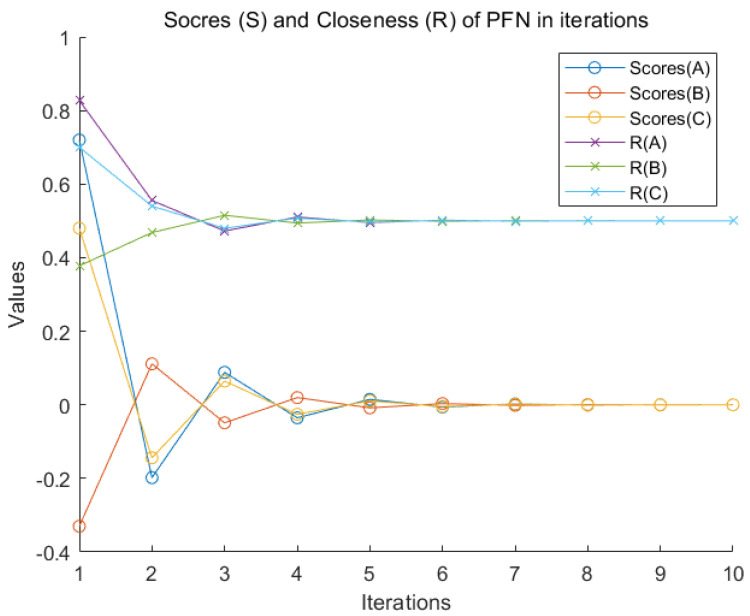
Score (S) and Closeness (*R*) of PFN negation.

**Figure 5 entropy-22-00195-f005:**
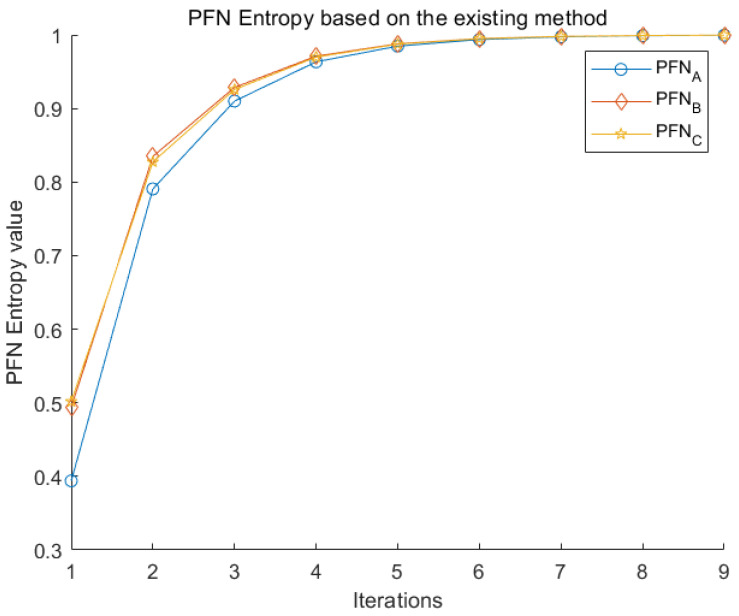
PFN uncertainty measure based on the existing method.

**Figure 6 entropy-22-00195-f006:**
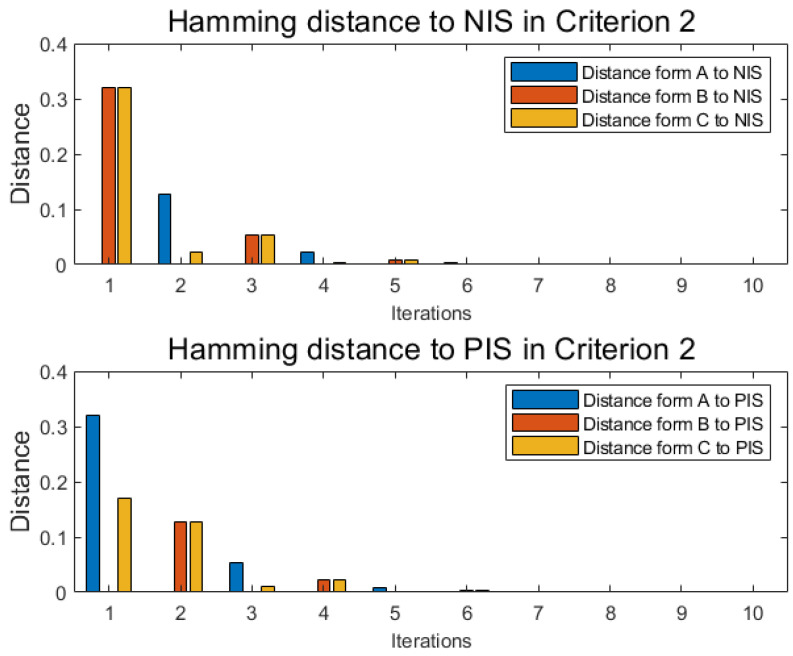
Distance in criterion 2.

**Figure 7 entropy-22-00195-f007:**
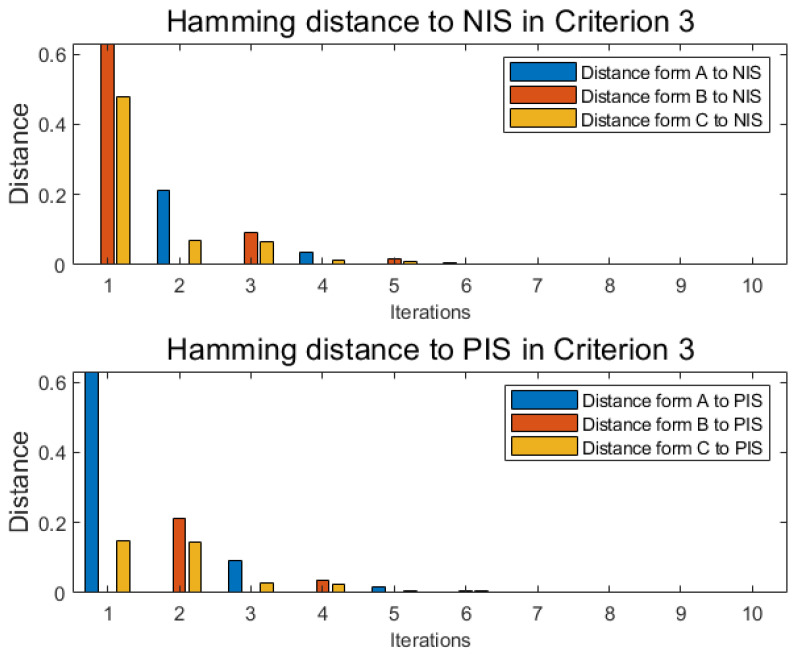
Distance in criterion 3.

**Figure 8 entropy-22-00195-f008:**
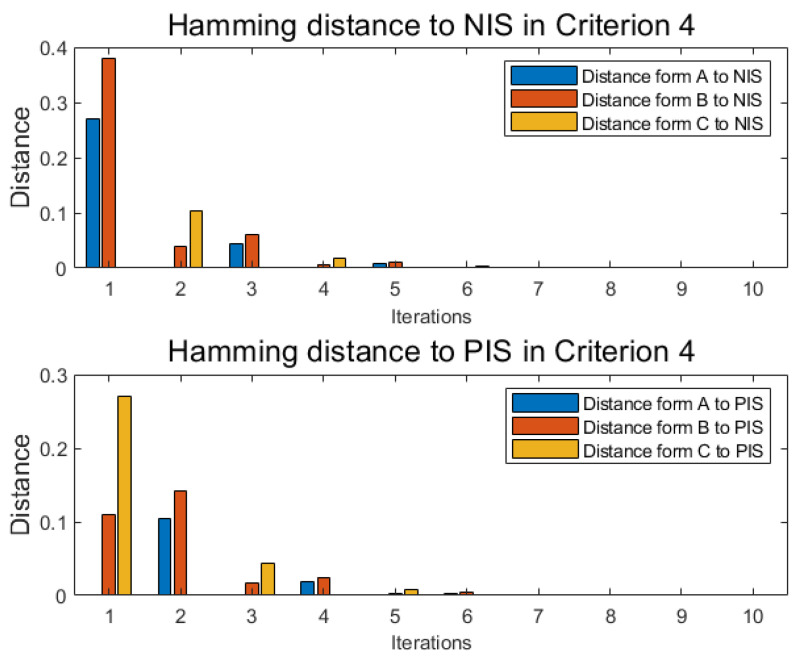
Distance in criterion 4.

**Figure 9 entropy-22-00195-f009:**
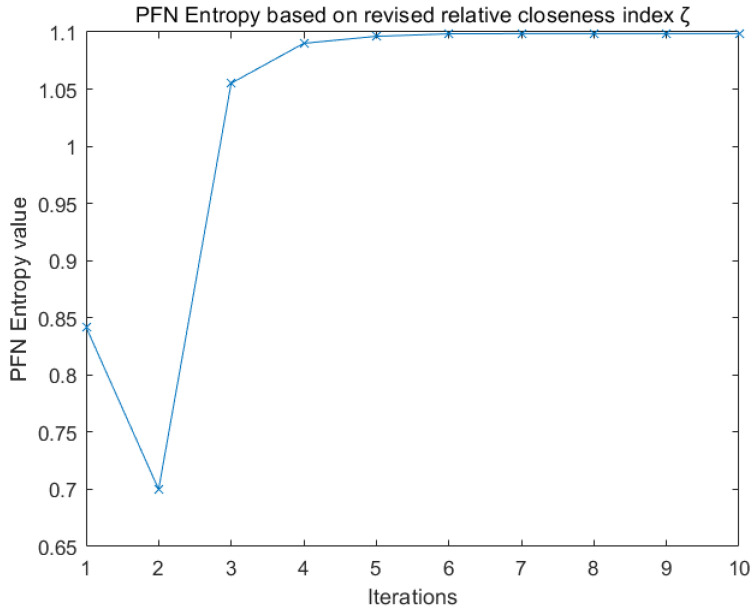
PFN Entropy.

**Figure 10 entropy-22-00195-f010:**
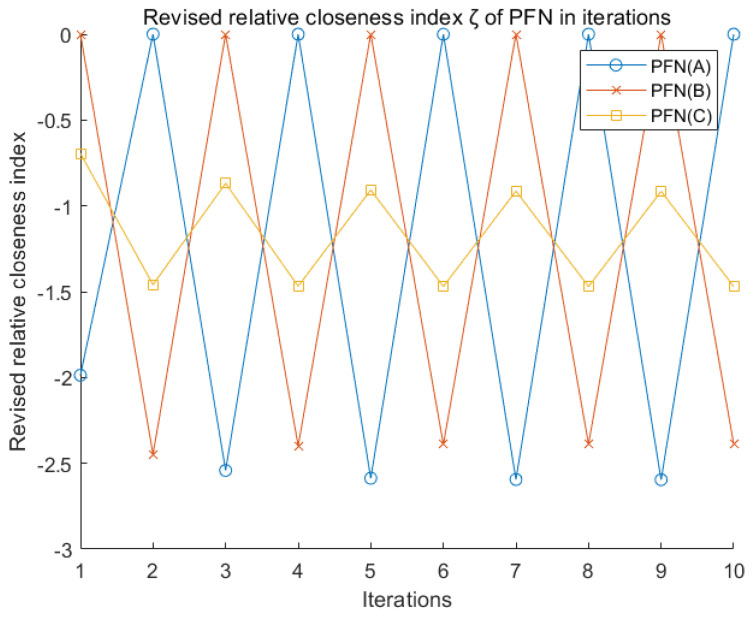
PFN revised relative closeness index (ζ) in MCDM.

**Table 1 entropy-22-00195-t001:** Single-criterion decision-making in PFN.

Alternative	Form of PFN	PFN
A	$\left(u_{A},\;v_{A}\right)$	(0.9,0.3)
B	$\left(u_{B},\;v_{B}\right)$	(0.4,0.7)
C	$\left(u_{C},\;v_{C}\right)$	(0.8,0.4)

**Table 2 entropy-22-00195-t002:** PFN negation and characters of alternative $x_{A}$.

Iteration	$u_{A}$	$v_{A}$	$\pi_{A}^{2}$	Score (S)	Closeness (*R*)
0	0.9000	0.3000	0.1000	0.7200	0.8273
1	0.0950	0.4550	0.7836	−0.1980	0.4445
2	0.4955	0.3965	0.5973	0.0883	0.5276
3	0.3772	0.4214	0.6801	−0.0353	0.4895
4	0.4288	0.4112	0.6470	0.0148	0.5045
5	0.4080	0.4155	0.6609	−0.0061	0.4982
6	0.4167	0.4137	0.6552	0.0025	0.5008
7	0.4132	0.4144	0.6575	−0.0011	0.4997
8	0.4146	0.4141	0.6566	0.0004	0.5001
9	0.4140	0.4142	0.6570	−0.0002	0.4999

**Table 3 entropy-22-00195-t003:** PFN negation and characters of alternative $x_{B}$.

Iteration	$u_{B}$	$v_{B}$	$\pi_{B}^{2}$	Score (S)	Closeness (*R*)
0	0.4000	0.7000	0.3500	−0.3300	0.3778
1	0.4200	0.2500	0.7586	0.1114	0.5317
2	0.4118	0.4675	0.6119	−0.0490	0.4848
3	0.4152	0.3907	0.6749	0.0197	0.5059
4	0.4138	0.4237	0.6493	−0.0083	0.4975
5	0.4144	0.4103	0.6600	0.0034	0.5010
6	0.4141	0.4158	0.6556	−0.0014	0.4996
7	0.4142	0.4135	0.6574	0.0006	0.5002
8	0.4142	0.4145	0.6566	−0.0002	0.4999
9	0.4142	0.4141	0.6569	0.0001	0.5000

**Table 4 entropy-22-00195-t004:** PFN negation and characters of alternative $x_{C}$.

Iteration	$u_{C}$	$v_{C}$	$\pi_{C}^{2}$	Score (S)	Closeness (*R*)
0	0.8000	0.4000	0.2000	0.4800	0.7000
1	0.1800	0.4200	0.7912	−0.1440	0.4598
2	0.4838	0.4118	0.5964	0.0645	0.5202
3	0.3830	0.4152	0.6809	−0.0257	0.4923
4	0.4267	0.4138	0.6467	0.0108	0.5033
5	0.4090	0.4144	0.6610	−0.0045	0.4987
6	0.4164	0.4141	0.6551	0.0018	0.5006
7	0.4133	0.4142	0.6576	−0.0008	0.4998
8	0.4146	0.4142	0.6566	0.0003	0.5001
9	0.4141	0.4142	0.6570	−0.0001	0.5000

**Table 5 entropy-22-00195-t005:** Scores function (S) of PFN.

Frequency of Iterations	S(A)	S(B)	S(C)	NIS	PIS
0	0.7200	−0.3300	0.4800	B	A
1	−0.1980	0.1114	−0.1440	A	B
2	0.0883	−0.0490	0.0645	B	A
3	−0.0353	0.0197	−0.0257	A	B
4	0.0148	−0.0083	0.0108	B	A
5	−0.0061	0.0034	−0.0045	A	B
6	0.0025	−0.0014	0.0018	B	A
7	−0.0011	0.0006	−0.0008	A	B
8	0.0004	−0.0002	0.0003	B	A
9	−0.0002	0.0001	−0.0001	A	B

**Table 6 entropy-22-00195-t006:** Distance to NIS $D(x_{i},x^{-})$.

Frequency of Iterations	$D(x_{A},x^{-})$	$D(x_{B},x^{-})$	$D(x_{C},x^{-})$
0	0.097500	0.000000	0.072000
1	0.000000	0.025106	0.004594
2	0.011389	0.000000	0.009672
3	0.000000	0.004513	0.000777
4	0.001901	0.000000	0.001622
5	0.000000	0.000782	0.000133
6	0.000325	0.000000	0.000277
7	0.000000	0.000134	0.000023
8	0.000056	0.000000	0.000048
9	0.000000	0.000023	0.000004

**Table 7 entropy-22-00195-t007:** Distance to PIS $D(x_{i},x^{+})$.

Frequency of Iterations	$D(x_{A},x^{-})$	$D(x_{B},x^{-})$	$D(x_{C},x^{-})$
0	0.000000	0.097500	0.025500
1	0.025106	0.000000	0.021600
2	0.000000	0.011389	0.001857
3	0.004513	0.000000	0.003860
4	0.000000	0.001901	0.000320
5	0.000782	0.000000	0.000668
6	0.000000	0.000325	0.000055
7	0.000134	0.000000	0.000115
8	0.000000	0.000056	0.000009
9	0.000023	0.000000	0.000020

**Table 8 entropy-22-00195-t008:** Relative closeness index (*R*) of PFN.

Frequency of Iterations	*R*(A)	*R*(B)	*R*(C)	NIS	PIS
0	0.8273	0.3778	0.7000	B	A
1	0.4445	0.5317	0.4598	A	B
2	0.5276	0.4848	0.5202	B	A
3	0.4895	0.5059	0.4923	A	B
4	0.5045	0.4975	0.5033	B	A
5	0.4982	0.5010	0.4987	A	B
6	0.5008	0.4996	0.5006	B	A
7	0.4997	0.5002	0.4998	A	B
8	0.5001	0.4999	0.5001	B	A
9	0.4999	0.5000	0.5000	A	B

**Table 9 entropy-22-00195-t009:** Multi-criteria decision-making in PFN.

Alternatives	Criterion 1	Criterion 2	Criterion 3	Criterion 4
$x_{A}$	(0.9, 0.3)	(0.7, 0.6)	(0.5, 0.8)	(0.6, 0.3)
$x_{B}$	(0.4, 0.7)	(0.9, 0.2)	(0.8, 0.1)	(0.5, 0.3)
$x_{C}$	(0.8, 0.4)	(0.8, 0.2)	(0.8, 0.4)	(0.6, 0.6)

**Table 10 entropy-22-00195-t010:** Distance $D(M,M^{C})$ with the existing method and total uncertainty.

Alternatives	Iteration	Criterion 1	Criterion 2	Criterion 3	Criterion 4	Total Uncertainty
$x_{A}$	1	0.1201	0.1706	0.2509	0.0644	0.3939
	2	0.0355	0.0631	0.0860	0.0248	0.7907
	3	0.0155	0.2070	0.0370	0.0105	0.9100
	4	0.0062	0.0110	0.0150	0.0043	0.9634
	5	0.0026	0.0046	0.0063	0.0018	0.9847
	6	0.0011	0.0019	0.0026	0.0007	0.9937
	7	0.0004	0.0008	0.0011	0.0003	0.9974
	8	0.0002	0.0003	0.0004	0.0001	0.9989
	9	0.0001	0.0001	0.0002	0.0001	0.9996
$x_{B}$	1	0.0637	0.2002	0.2127	0.0293	0.4941
	2	0.0230	0.0591	0.0706	0.0125	0.8348
	3	0.0099	0.0258	0.0306	0.0051	0.9286
	4	0.0040	0.0104	0.0124	0.0021	0.9711
	5	0.0017	0.0044	0.0052	0.0009	0.9879
	6	0.0007	0.0018	0.0021	0.0004	0.9950
	7	0.0003	0.0007	0.0009	0.0002	0.9979
	8	0.0001	0.0003	0.0004	0.0001	0.9991
	9	0.0000	0.0001	0.0002	0.0000	0.9996
$x_{C}$	1	0.0049	0.1519	0.2127	0.1288	0.5017
	2	0.0020	0.0504	0.0706	0.0495	0.8274
	3	0.0008	0.0218	0.0306	0.0210	0.9257
	4	0.0004	0.0088	0.0124	0.0086	0.9698
	5	0.0001	0.0037	0.0052	0.0036	0.9874
	6	0.0001	0.0015	0.0021	0.0015	0.9948
	7	0.0000	0.0006	0.0009	0.0006	0.9978
	8	0.0000	0.0003	0.0004	0.0003	0.9991
	9	0.0000	0.0001	0.0002	0.0001	0.9996

**Table 11 entropy-22-00195-t011:** Distance to NIS $D(x_{i},x^{-})$ after PFWA.

Iterations	$D(x_{A},x^{-})$	$D(x_{B},x^{-})$	$D(x_{C},x^{-})$
0	0.165000	0.395500	0.320000
1	0.106419	0.034663	0.060613
2	0.022444	0.060638	0.045579
3	0.018510	0.006206	0.010400
4	0.003787	0.010306	0.007690
5	0.003186	0.001074	0.001786
6	0.000648	0.001766	0.001316
7	0.000547	0.000185	0.000306
8	0.000111	0.000303	0.000226
9	0.000094	0.000032	0.000053

**Table 12 entropy-22-00195-t012:** Distance to PIS $D(x_{i},x^{+})$ after PFWA.

Iterations	$D(x_{A},x^{+})$	$D(x_{B},x^{+})$	$D(x_{C},x^{+})$
0	0.300500	0.125000	0.188000
1	0.051263	0.142131	0.104000
2	0.045386	0.015587	0.025252
3	0.009049	0.024740	0.018371
4	0.007712	0.002610	0.004315
5	0.001562	0.004259	0.003171
6	0.001321	0.000446	0.000740
7	0.000268	0.000731	0.000545
8	0.000227	0.000077	0.000127
9	0.000046	0.000125	0.000094

**Table 13 entropy-22-00195-t013:** Revised relative closeness index (ζ) and PFN entropy ($E_{\zeta }$).

Iterations	ζ(A)	ζ(B)	ζ(C)	Comparison	$E_{\zeta }$
0	−1.9868	0.0000	−0.6949	A ≺ C ≺ B	0.841884
1	0.0000	−2.4469	−1.4592	B ≺ C ≺ A	0.699963
2	−2.5417	0.0000	−0.8684	A ≺ C ≺ B	1.055133
3	0.0000	−2.3987	−1.4683	B ≺ C ≺ A	1.090053
4	−2.5869	0.0000	−0.9070	A ≺ C ≺ B	1.096197
5	0.0000	−2.3896	−1.4698	B ≺ C ≺ A	1.098365
6	−2.5942	0.0000	−0.9135	A ≺ C ≺ B	1.098531
7	0.0000	−2.3880	−1.4700	B ≺ C ≺ A	1.098605
8	−2.5955	0.0000	−0.9146	A ≺ C ≺ B	1.098610
9	0.0000	−2.3877	−1.4701	B ≺ C ≺ A	1.098612

**Table 14 entropy-22-00195-t014:** Revised relative closeness index (ζ) of PFN in numerical example 1.

Frequency of Iterations	ζ(A)	ζ(B)	ζ(C)
0	NaN	−∞	−∞
1	−∞	NaN	−∞
2	NaN	−∞	−∞
3	−∞	NaN	−∞
4	NaN	−∞	−∞
5	−∞	NaN	−∞
6	NaN	−∞	−∞
7	−∞	NaN	−∞
8	NaN	−∞	−∞
9	−∞	NaN	−∞

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
