# Peer review of "Negation of Pythagorean Fuzzy Number Based on a New Uncertainty Measure Applied in a Service Supplier Selection System"

_entropy, 2020, doi:10.3390/e22020195_

Round 1

Reviewer 1 Report

The goal of this paper is to propose a new entropy of Pythagorean fuzzy number based on a TOPSIS method. The proposed method is interesting.

The following comments are offered to improve the quality of this paper.

The title can be reorganised. The title of this paper can be revised to make it more attractive to readers.

As the title suggests that the new method is proposed for supporting the selection process involving service supplier selection system, it is important for the authors to detail the scenario. In number example 2, the authors have mentioned three major airlines. I would suggest providing more details about these alternatives to strengthen their example.

The heading for section two can be changed from “Preliminaries” to “Literature review” or something suitable.

Similarly, the heading “Further future research” in the conclusion section can be changed to “Future research” or “Future research opportunities”.

The authors have made significant effort in detailing the steps involved in the selection process. As there are steps involved int the selection process, it would be a good idea to present a diagram summarising the steps involved in the process.

There paper needs to be proof-read.

The conclusion section needs to include limitations and research implications.  

Overall, the authors have made significant contributions by presenting a strong case for the proposed method and testing its suitability.

I recommend “Accept with minor revision”.

Author Response

Dear reviewer:

Thank you for your constructive comments. The answer was written in Word. 

best wishes!

Corresponding author: Cai

Reviewer 2 Report

The article is quite interesting, but needs to be corrected before adoption.

1. Orcid number is not given (only logo);
2. the summary should be improved so that it fully illustrates what is the main contribution of the work and what were the motivations for undertaking this study. In short, a good summary should provide a brief background; motivation, justification and contribution of the work; main results and conclusions. Currently, the abstract is quite chaotic.
3. the first paragraph of the first section has half the literature references used throughout the work. The vast majority of them are insignificant. Instead, a separate satnified overview of tools/methods for dealing with uncertainty should be written.
4. The literature references in the second paragraph are incorrect e.g. [47] and [48] do not refer to PFN.
5. The introduction should clearly and precisely define the whole contribution of the presented paper together with the justification of the necessity to undertake the research (identification of l=uki).
6. from line 102 the TOPSIS method, which has not been introduced anywhere, begins to appear. The potential user would be able to refer to the significance of the contribution from this study. In addition, other studies of the TOPSIS method dealing with uncertainty should be reviewed (another missing literature review and definition).
7 The algorithm is presented as 4 functions. It should be separated for more readability and/or added to the flowchart.
8. numerical results should be compared with existing solutions to show the superiority of the proposed method over existing ones
9 Drawings have a large degree of pixelosis. This must be corrected
10. the Conclusions must refer to the amendments made to the work, especially points 6 and 8.

Author Response

(The authors gave the same response as above.)

Reviewer 3 Report

The contribution is appealing and timely. It allows to define a new form of entropy. This parameter could be good for handling uncertainty. Moreover more topics regarding the problems are discussed in the paper. 

The methods are new in my knowledge. 

The introduction of the paper is very accurate and the  preliminary concepts are well expressed in order to understand the main part of the paper. I view the presented approach in perspective very useful in the application of fuzzy computation by using microfluidics based hardware.

I suggest the authors to view the following paper to have ideas for their future research and express some comments in the conclusion of the paper

Anandan, P., Gagliano, S., Bucolo, M. Computational models in microfluidic bubble logic Microfluidics and Nanofluidics, Volume 18, Issue 2, 2014, Pages 305-321   I suggest to include it in the references. In my opinion the joint of innovative ideas to handle information in new algorithms with the possibility of using non traditional hardware should be very useful to design specialized and new class of computational devices.    Moreover the suitable definition of fuzzy set allows the modelling of innovative systems.   I hope in the joint of the concepts to improve the science in a future vision. It is important and I invite the authors of many of my reviews to approach the problems in an interdisciplinary visionary manner.   Moreover the introduction of new ideas and to introduce briefly some possible concept make the paper more appealing to the  reader and the journal more competitive. Welcome!

Author Response

(The authors gave the same response as above.)

Round 2

Reviewer 2 Report

The authors positively responded to most of my comments. The manuscript has been improved to such an extent that it can be accepted as it stands.